# Plasma Cell-Free DNA as a Novel Biomarker for the Diagnosis and Monitoring of Atherosclerosis

**DOI:** 10.3390/cells11203248

**Published:** 2022-10-16

**Authors:** Benheng Qian, Kexin Li, Xiaoying Lou, Ye Guo, Yidong Wang, Lianpin Wu, Donghong Zhang

**Affiliations:** 1Department of Cardiology, The Second Affiliated Hospital of Wenzhou Medical University, Wenzhou 325027, China; 2State Key Laboratory of Molecular Oncology, Department of Clinical Laboratory, National Cancer Center/National Clinical Research Center for Cancer/Cancer Hospital, Chinese Academy of Medical Sciences and Peking Union Medical College, Beijing 100021, China; 3Department of Laboratory Medicine, Peking Union Medical College Hospital and Peking Union Medical College, Beijing 100021, China; 4The Institute of Cardiovascular Sciences, School of Basic Medical Sciences, Xi’an Jiaotong University Health Science Center, Xi’an 710049, China

**Keywords:** atherosclerosis, cell-free DNA, diagnosis, progression, new biomarker

## Abstract

Atherosclerosis (AS) is the leading cause of cardiovascular diseases (CVDs) with a high rate of mortality worldwide. Plasma cell-free DNA (cfDNA), mainly originating from apoptosis, necrosis, and active secretion, has been recognized as a promising biomarker for the diagnosis and prognosis of multiple cancers, whereas there are no reports about cfDNA in CVDs. Here, we found an increased quantity and decreased integrity of cfDNA (cfDI) in the serum from AS patients compared with normal controls. Moreover, the reduced cfDI is inversely correlated with serum LDL levels, carotid plaque size, and carotid plaque thickness in the progression of AS. Consistently, in vivo experiments confirmed that the release and cleavage of cfDNA were increased concomitantly with the development and progression of AS in ApoE^−^/^−^ mice. Our study sheds light on the potential of cfDNA and cfDI as molecular biomarkers for detecting and monitoring AS.

## 1. Introduction

Circulating cell-free DNA (cfDNA) mainly enter the peripheral blood upon apoptosis, necrosis, and active secretion, as part of regular cell turnover or pathology [1]. In recent years, this understanding has led to emerging studies on cfDNA as a diagnostic tool, impacting multiple areas of medicine [2,3,4]. Of these pieces of research, one promising approach is to analyze the distribution of DNA fragments of different lengths. As the size of plasma DNA can be attributed to the caspase-dependent cleavage, nucleosomal organization and chromatin accessibility [5], the fragmentation profiles of cfDNA are involved with gene expression and multiple other cellular processes [6]. Therefore, the DNA integrity of cfDNA (cfDI), which is a measure of the extent of cfDNA fragmentation, has been exploited as a biomarker for diagnosis and prognostication in multiple diseases [7,8,9]. This is because the index of cfDNA and cfDI, DNA fragments ALU (ALU 115, short fragment and ALU 247, long fragment), and *Line1* (*Line1* OFR2, short fragment and *Line1* 5′UTR, long fragment) have been explored in most diseases, such as infectious diseases, autoimmune disorders, myocardial infarction, and multiple cancers [10,11,12,13]. Specifically, the ratio of ALU 115 and ALU 247 were commonly used as cfDI for human targets [7,14,15], while *Line1* OFR2 and *Line1* 5′UTR were more often used in mouse [16,17].

Atherosclerosis (AS) is a chronic arterial disease and the most frequent underlying cause of vascular disease worldwide [18,19]. With the rupture or erosion of plaque, atherosclerosis precipitates life-threatening clinical events, such as acute coronary syndromes and ischemic stroke [20]. The assessment of plaque burden is commonly achieved by evaluating the progression of carotid intima media thickness (CIMT) [21,22]. To prevent AS and ensuing life-threatening diseases, we must strive to develop new diagnostic methods beyond B-mode ultrasonography or angiography, which could not visualize minimal atherosclerosis lesions throughout the body [23,24].

The emergence of highly sensitive cfDNA assays has shown promising clinical potential as a way to detect the molecular residual disease (MRD) for solid tumors [25,26,27]. Moreover, most of the cell-free DNA molecules in plasma are derived from white blood cells and vascular endothelial cells in healthy persons [28], which are the major players in the development of atherosclerosis. On the other hand, a growing number of studies support the occurrence of the programmed cell death of vascular smooth muscle cells and endothelial cells in the progression of atherosclerosis [29,30]. Further research even proved that neutrophil extracellular traps (NETs), which are decondensed chromatins expelled from suicidal neutrophils, could be detected in atherosclerotic lesions [31,32]. Collectively, these findings have made cfDNA a promising biomarker in detecting minimal atherosclerotic lesions instead of B-mode ultrasonography or angiography.

In this study, we determine the profile of cfDNA and cfDI in different stages of atherosclerosis. We found that patients with carotid plaque possess an elevated serum cfDNA, as well as a decreased cfDI level compared with healthy individuals. Moreover, cfDI was negatively correlated with serum LDL levels and carotid plaque size and thickness. Using two different animal models, we further certified that both the release and cleavage of cfDNA were increased in the progression of plaque. Thus, both cfDNA and cfDI could be sensitive markers in predicting the progression of plaque.

## 2. Materials and Methods

### 2.1. Study Cohort

The study was approved by the Ethical Committee of the Second Affiliated Hospital of Wenzhou Medical University. All candidates were enrolled with written informed consent. For the normal group, we randomly selected 113 sex- and age-matched healthy individuals from patients undergoing a health examination. The AS cohort consisted of patients with at least one atherosclerotic lesion in the aorta artery, carotid artery, cerebrovascular region, coronary artery, or kidney artery. The diagnosis was confirmed by CT arteriography, aortic angiography, or MRI. Patients with myocardial infarction, stroke, coronary revascularization, or peripheral vascular surgery during the preceding 6 months were excluded. We also excluded patients with current infections, genetic diabetes, uremia, active malignancies, aneurysm, inflammatory diseases, or iatrogenic and traumatic aortic dissections. Detailed patient characteristics are shown in Appendix A.

### 2.2. Sample Preparation and Extraction of cfDNA

A total of 4 mL of fasting cubital venous blood was collected from each participant at the time of diagnosis before any therapy or surgery. To separate the plasma, blood was centrifuged at 1500× *g* for 5 min at 4 °C within 2 h of blood collection. The supernatant was further centrifuged at 15,000× *g* for 10 min at 4 °C to minimize any blood cell or cell debris contamination. cfDNA was then extracted from 200 µL of serum using Serum/Plasma Circulating DNA Kit (Tiangen, Beijing, China) following the manufacturer’s instructions. The resulting cfDNA was stored at −80 °C for further use.

### 2.3. Estimation of DNA Integrity and Concentration of cfDNA

The DNA integrity and concentration of cfDNA were derived by an analysis with two repetitive elements. For human targets, a short fragment (ALU 115) and a long fragment (ALU 247) were, respectively, amplified and quantified. As for mouse targets, we used *Line1* 5′UTR (amplicon size: 247 bp) as the long fragment and *Line1* OFR2 (amplicon size: 170 bp) as the short fragment. All primers were certified by previous researchers [14,17,33]. Each assay was carried out in triplicate with the LightCycler 480II system (Roche Rotkreuz, Rotkreuz, Switzerland). The absolute amount of serum DNA fragments in each sample was determined using a standard curve with serial dilutions. DNA integrity index was subsequently calculated as a ratio of the concentration of long fragment to the concentration of short fragment. To avoid inter-run variations, samples from different groups were randomized prior to qPCR setup, and the assays were conducted in a blinded manner.

### 2.4. Laboratory Analyses

All blood samples were obtained early in the morning following 8 h of overnight fasting. Biochemical variables, including the serum concentrations of triglyceride (TG), total cholesterol (TC), high-density lipoprotein cholesterol (HDL), low-density lipoprotein cholesterol (LDL), apolipoprotein A1 (ApoA1), apolipoprotein B (ApoB), folate, and vitamin B12 (VB12) were determined by routine techniques using an automated analyzer (Beckman AU5800) at the Clinical Laboratory Department of the Second Affiliated Hospital of Wenzhou Medical University. The coefficients of variation were <10% for homocysteine, hs-CRP (high-sensitivity C-reactive protein), and folate, and <5% for all other variables.

### 2.5. CIMT Measurements

The carotid intima media thickness (CIMT) of both the left and right carotid arteries were measured according to a standardized protocol using a B-mode ultrasonic diagnostic apparatus (GE Healthcare, Chicago, IL, USA). Specifically, the participants were placed in the supine position with their heads turned at an angle of 45° towards the opposite direction of the ultrasound operator; bilateral carotid arteries were then scanned to obtain technical data, including the CIMT. The CIMT was determined as the measured distance between the luminal–intimal interface and the media–adventitial interface. A CIMT of 1.0–1.3 mm was defined as carotid AS; >1.3 mm, 0.5 mm thicker than the adjacent site, or >1.5 times that of the adjacent site was considered to have a presence of carotid plaques. Each measurement was blindly repeated 5–8 times by two different sonographers, and the means of the three highest values were recorded and used for statistics. The intra-observer variability was estimated to be below 3%.

### 2.6. Animal Models

Male apolipoprotein E-knockout (ApoE^−^/^−^) C57BL/6J mice were purchased from the Nanjing Biomedical Research Institute of Nanjing University. All animal procedures and experiments were performed under guidelines approved by the animal studies committee of Wenzhou Medical University. The mice were bred 3–5 mice per cage in a temperature and humidity-controlled environment with a reverse 12-h light/dark cycle. Eight-week-old male ApoE^−^/^−^ mice were randomized in different experiment groups and blindingly fed a normal diet (ND) or western diet (WD) for 2 and 6 months. The composition of normal chow diet is based on the NIH feeds standard (NIH-07). The western diet was obtained from the Research Diets (D12079B, New Brunswick, NJ, USA), containing 17 kcal% protein, 43% carbohydrate, and 41 kcal% fat. At multiple time points, the mice were anesthetized by an intraperitoneal injection of pentobarbital (50 mg/kg bodyweight). The foot reflex was monitored to assess the degree of anesthesia. About 1 mL of blood was then collected and separated into serum and leukocytes. Subsequently, the mice were perfused with 10 mL of PBS for washing blood and with 4% paraformaldehyde (PFA) for prefixing. The whole aortas, including aortic root, thoracic aorta, and abdominal aorta, were then isolated and further fixed in 4% PFA at 4 °C for 24 h. Then, the whole aortas were embedded in optimal cutting temperature compounding with orientation for front sections (5 μm) and HE staining.

### 2.7. Statistical Analysis

All statistical analyses were performed using SPSS 22.0 (SPSS Inc., Chicago, IL, USA). The data are presented as mean ± standard deviation (SD). A Student’s *t* test (unpaired) was performed to compare differences between the two groups. A comparison of three or more groups was performed by an ANOVA analysis. The correlation of continuous variables was examined by the Spearman correlation assay. A linear and a multivariable regression analysis were used to examine the association of clinical characteristics and cfDNA or cfDI with or without adjustment for age and sex. The results were considered statistically significant at *p* < 0.05 (two tailed). Statistical significances were classified as * *p* < 0.05; ** *p* < 0.01; *** *p* < 0.001, **** *p* < 0.0001.

## 3. Results

### 3.1. Diagnostic Role of cfDNA in Human AS Patients

To test the diagnostic role of cfDNA in AS patients, the cfDNA concentrations of long (ALU 247) and short fragments (ALU 115) were measured from 102 cases of AS patients and 113 cases of normal controls. Interestingly, both the long fragments and short fragments of cfDNA were increased in AS patients (Figure 1A,B), while decreased cfDI (ALU 247/ALU 115) was found in AS patients compared with the controls (Figure 1C). In addition, no correlations were found between age and cfDI in both AS patients and the controls (Appendix A). There was no association between cfDI and gender in AS patients; on the other hand, decreased cfDI levels were found in males in the control group (Appendix A). Consistently, the ROC assay showed that all three features of cfDNA could distinguish AS patients from normal controls. Notably, cfDI showed more diagnostic potential for AS patients (AUC = 0.780, *p* < 0.0001). These results suggest that the release and cleavage of cfDNA could act as a potential diagnostic biomarker for AS.

### 3.2. Decrease in cfDI Is Associated with Plaque Progression

The Spearman association between dynamic cfDI level and AS progression was further explored. As displayed in Figure 2A,B, cfDI showed a mild but significant negative correlation with both carotid plaque size (r = −0.285, *p* = 0.004) and thickness (r = −0.333, *p* = 0.001). We then classified the AS patients into Class-I, -II and -III groups according to their plaque levels and found that AS patients with the largest or thickest carotid plaque had the lowest cfDI level (Figure 2C,D). After adjusting for age and sex, a linear regression analysis further confirmed that cfDI was negatively associated with the plaque size and plaque thickness (Table 1).

We next assessed the potential association between the cfDI and biochemical characteristics of AS patients. A significant negative correlation was found between cfDI and serum LDL levels (r = −0.236, *p* = 0.017) in AS patients, but not in the normal group (Appendix A). This result was further validated by the linear regression analysis adjusted by age and sex (Table 1). Additionally, cfDI was negatively related with serum apolipoprotein A1 (ApoA1) and homocysteine (Hcy) levels (Table 1), which also play major roles in the development of AS [34,35]. Collectively, the release and cleavage of cfDNA might correlate with LDL, ApoA1, and Hcy, and might be involved in the plaque progression of AS patients.

### 3.3. Release and Cleavage of cfDNA Were Gradually Increased during the Development of AS in ApoE^−^/^−^ Mice

To confirm the above observation, we detected the concentration and integrity of cfDNA in apolipoprotein E knockout mice (ApoE^−^/^−^) that spontaneously developed AS. Similarly, both the long (*Line1* 5′UTR) and short (*Line1* ORF2) fragments of cfDNA were increased during the aging of ApoE^−^/^−^ mice. Significantly elevated levels of cfDNA were found after 12 months, which were also accompanied by the reduction in cfDI (*Line1* 5′UTR/*Line1* ORF2) (Figure 3A–C). To exclude the influence of aging on our observations [36,37], we further generated a mouse aortic AS model induced by western diet (WD) food in males at 2 months. After 6 months of WD treatment, the mice developed severe plaque burden in the aortic root compared with mice fed a normal diet (ND) (Figure 4A,B). Interestingly, the elevated levels of cfDNA were expedited in accordance with the progression of plaque at 6 months (Figure 3D,E). Concomitantly, a reduction in cfDI was observed in WD-induced AS mice (Figure 3F), suggesting that the release and cleavage of cfDNA was increased with the development of AS.

### 3.4. Release and Cleavage of cfDNA Were Associated with Plaque Progression In Vivo

To further reveal the correlation between cfDNA and AS progression, we next compared the cfDNA among different plaque levels in WD-induced AS mice. Interestingly, the size of carotid plaque was positivity correlated with both long (r = 0.511, *p* = 0.0002) and short fragments (r = 0.443, *p* = 0.001) (Figure 4B) and negatively correlated with cfDI (r = −0.339, *p* = 0.041). Similar correlation patterns were observed in cfDNA and plaque thickness (Figure 4C). We further classified the mice into four different groups according to their plaque size. Consistently, both the short and long fragments of cfDNA were increased with the plaque size (Figure 4D,E). On the other hand, the cfDI showed significantly lower levels in mice with the largest carotid plaque (Figure 4F). Collectively, our results demonstrate that the release and cleavage of cfDNA are closely associated with the progression of AS.

## 4. Discussion

In this study, we first found that cfDNA was increased in the serum of AS patients, while the integrity of cfDNA was decreased. Moreover, cfDI was negatively correlated with serum LDL levels, ApoA1, HCY, carotid plaque size, and carotid plaque thickness. Our animal experiments further confirmed that the progression of AS contributed to the dynamic loss of cfDI. Consequently, our study suggested that the progression of atherosclerosis was accompanied by the increased release and cleavage of cfDNA.

Cell-free DNA (cfDNA) are DNA fragments of variable length between 50 and 250 base pairs, which dissociated in the plasma [38]. cfDNA has become an attractive research subject as a non-invasive biomarker in diseases. As the index of cfDNA and cfDI, DNA fragments ALU 115 and ALU 247 have been explored in most diseases, such as infectious diseases, autoimmune disorders, myocardial infarction, and multiple cancers [10,11,12,13]. In our current study, we firstly found that cfDI could act as a novel biomarker during AS development and progression. Overall, our result and previous evidence indicated that cfDI could be used as a potential independent biomarker in most invasive diagnostics, improving disease monitoring, clinical decision making, and patients’ outcome in the future.

Atherosclerosis is a multifocal, smoldering, immunoinflammatory disease which cannot be detected until it involves the superficial large arteries and, therefore, can be visualized by B-mode ultrasonography [18,39]. Notably, cfDNA and cfDI have become a promising biomarker for detecting minimal residual disease in multiple tumors [25,26,27]. Moreover, most of the cfDNA molecules in plasma are derived from white blood cells (55%) and vascular endothelial cells (10%) in healthy individuals [28], which suggests that cfDNA might be a promising biomarker in atherosclerosis. In our study, we found a significant difference in cfDNA and cfDI levels between AS patients and normal individuals. The ROC curves further showed cfDNA and cfDI as good biomarkers for diagnosing AS. Moreover, the inverse correlation between cfDI and plaque progression indicated the potential of cfDNA in monitoring plaque burden. Significant advances have been made in the novel biomarker identification for AS development and progression. A developed logistic model based on the well-known AS risk factors such as gender, age, hypertension, and diabetes shows excellent predictive ability (AUC = 0.808) and might have a significant clinical implication for detecting subclinical atherosclerosis in patients with chronic kidney disease [40]. Additionally, circulating miRNA-933 has proven to be related with hyperlipidemia (AUC = 0.739) and atherosclerosis (AUC = 0.703) [41]. A circulating protein panel including IGHA2, APOA, and HPT was also identified to predict AS development (AUC = 0.73) [42]. Therefore, the cfDI (AUC = 0.780) identified in our study has a comparable diagnostic value as the above biomarkers for AS patients.

Circulating cell-free DNA enters the peripheral blood upon cell death as a part of normal cell turnover or pathology. Additionally, emerging research observed the increased apoptosis of macrophages, smooth muscle cells, and endothelial cells during the progression of AS [43]. Studies show that extracellular vesicles derived from LDL-induced endothelial cell apoptosis could contribute to plaque progression via the transfer of cellular contents such as cleaved nucleic acids [30]. Meanwhile, intraplaque macrophages, another major player in AS, are found to undergo apoptosis during all stages of atherosclerosis [44]. In this article, we found increased cfDNA as well as decreased cfDI in the AS serum. Since most of the cfDNA originates from white blood cells and vascular endothelial cells in normal circumstances, it is conceivable that the dysregulation of apoptosis in AS lesions could lead to the increased release and cleavage of cfDNA that correlates with the progression of plaque, thus, making cfDNA and cfDI suitable biomarkers for the early detection and monitoring of atherosclerosis.

Multiple studies have proved that cfDNA may have different epigenetic modifications depending on its origin [45,46,47]. By analyzing the DNA methylation profiles of cfDNA, researchers can determine which cell releases cfDNA under different pathological conditions. In this study, we found an increased release and cleavage of cfDNA during the progression of atherosclerosis. Future studies should focus on finding the specific cfDNA fragments released in AS using multiple genomics sequencing tools.

In summary, our clinical investigation observed an increased concentration as well as a decreased integrity of cfDNA in the serum of AS patients. Moreover, reduced cfDI was found to be inversely correlated with serum LDL levels, carotid plaque size, and carotid plaque thickness. Subsequent studies in two different animal models further confirmed that the release and cleavage of cell-free DNA occurred during the development and progression of atherosclerosis. Our results point to the potential of cfDNA and cfDI as molecular biomarkers for diagnosing and monitoring AS.

## Figures and Tables

**Figure 1 cells-11-03248-f001:**
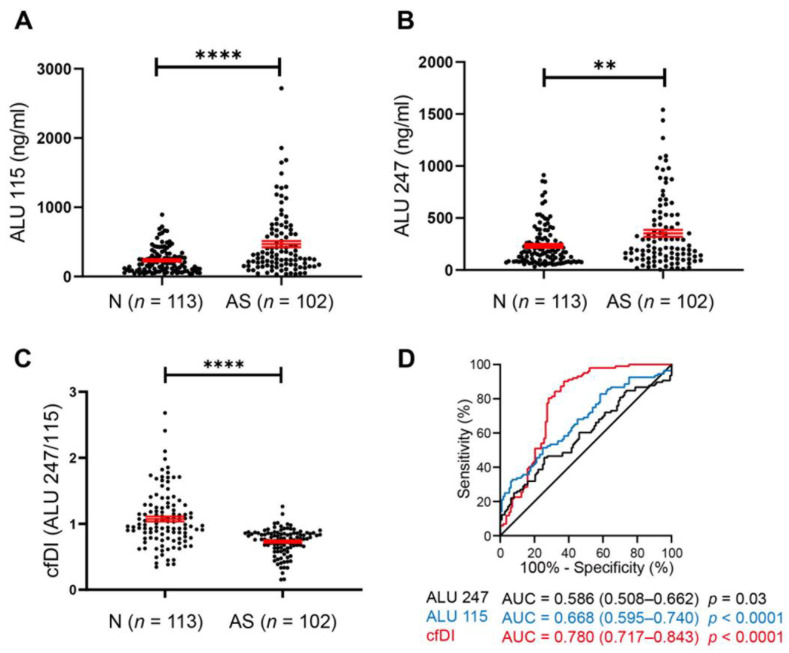
cfDNA acts as a new biomarker in atherosclerosis (AS). (**A**,**B**) The cfDNA concentration of short fragments (ALU 115) (**A**) and long fragments (ALU 247) (B) in AS patients and normal individuals. (**C**) The level of cfDI (cfDNA integrity, ALU 247/ALU 115) in patient group and control group. (**D**) ROC curves of ALU 115, ALU 247 and cfDI. Data are shown as mean ± SEM and were evaluated by unpaired *t* test. ** *p* < 0.01, **** *p* < 0.0001.

**Figure 2 cells-11-03248-f002:**
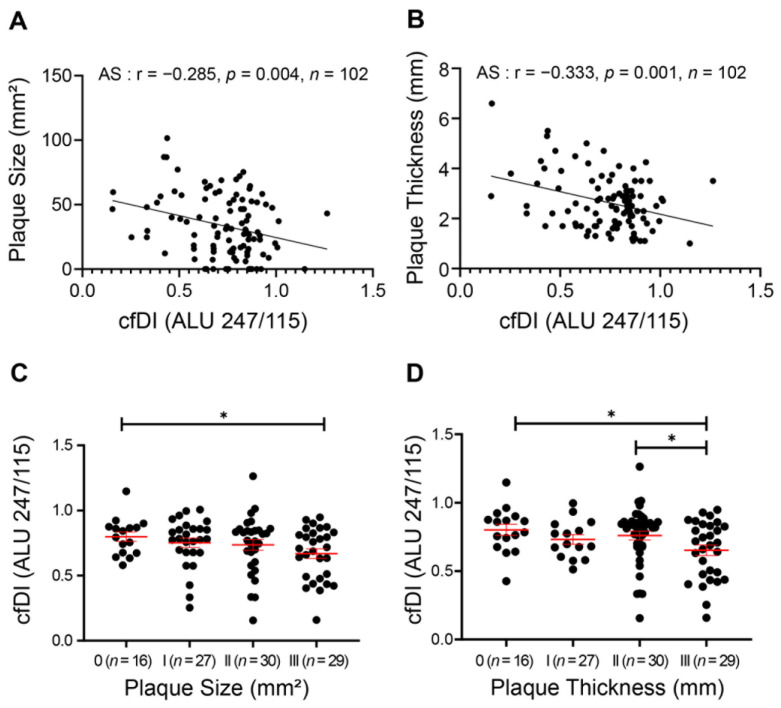
Reduced cfDI associates with the carotid plaque progression in atherosclerosis. (**A**,**B**) Spearman correlation coefficients for cfDI correlated with carotid plaque size (**A**), and carotid intima media thickness (CIMT) (**B**). (**C**,**D**) Levels of cfDI among different groups of carotid plaque size (**C**) and CIMT (**D**). Data are mean ± SEM. * *p* < 0.05.

**Figure 3 cells-11-03248-f003:**
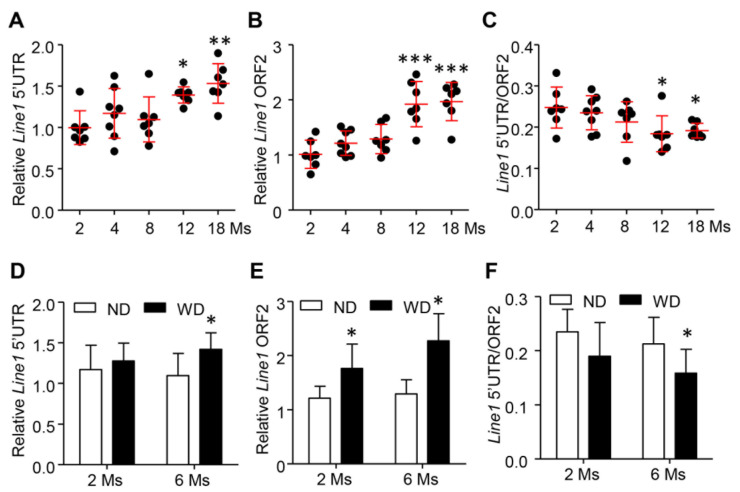
Release and cleavage of cfDNA were increased during the development of AS in ApoE^−^/^−^ mice. (**A**,**B**), The cfDNA concentration of long fragment (*Line1* 5′UTR) (**A**) and short fragment (*Line1* ORF2) (**B**) in the serum of ApoE^−^/^−^ mice at different stages. (**C**) Levels of cfDI (*Line1* 5′UTR/*Line1* ORF2) during the development of ApoE^−^/^−^ mice. (**D**,**E**) The concentration of long fragment (**D**) and short fragment (**E**) of cfDNA between ApoE^−^/^−^ mice fed with 6 months of western diet (WD) and normal diet (ND). (**F**) Levels of cfDI in ApoE^−^/^−^ mice fed with WD and ND. Data are shown as mean ± SEM and were evaluated by unpaired *t* test. * *p* < 0.05, ** *p* < 0.01, *** *p* < 0.001.

**Figure 4 cells-11-03248-f004:**
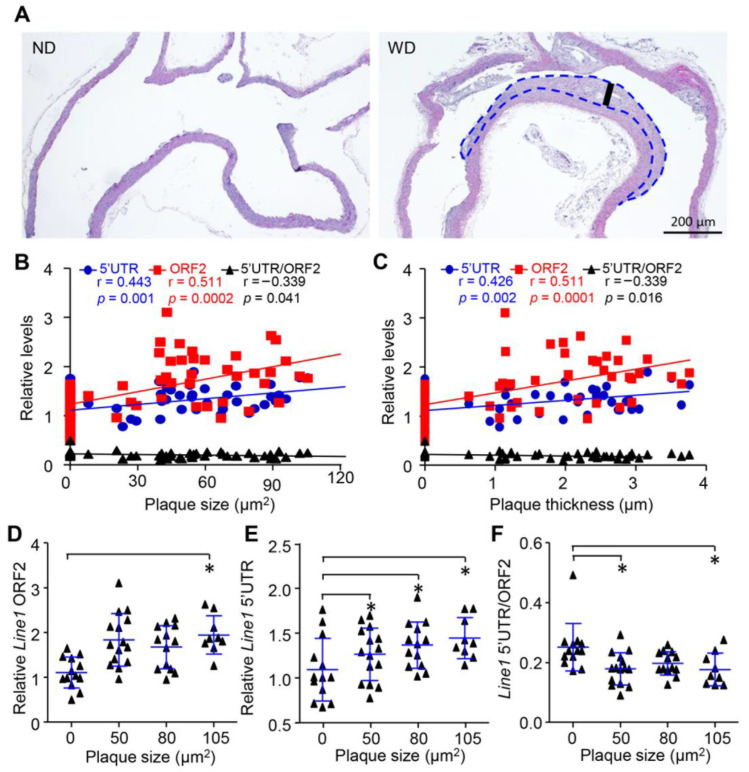
Release and cleavage of cfDNA was associated with plaque progression in vivo. (**A**), HE staining of aorta from ApoE^−^/^−^ mice fed with WD and ND for 6 months. (**B**,**C**) Spearman correlation for cfDNA concentration correlated with carotid plaque size (B) and thickness (**C**). (**D**,**F**) Levels of short fragment (**D**), long fragment (**E**) and integrity (**F**) of cfDNA among different levels of plaque size. Data are shown as mean ± SEM and were evaluated by unpaired *t* test. * *p* < 0.05.

**Table 1 cells-11-03248-t001:** Linear regression analysis of the association between cfDI and clinical factors adjusted by age and gender.

Clinical Factors	AS (*n* = 102)					N (*n* = 113)				
	β	*p* Value	95% CI			β	*p* Value	95% CI		
Plaque size (mm^2^)	−2.784	0.006	−0.004	to	−0.001					
Plaque thickness (mm)	−3.306	0.001	−0.101	to	−0.025					
CRP (mg/dL)	−0.018	0.985	−0.002	to	0.002	−0.106	0.916	−0.035	to	0.031
WBC (×10^9^/L)	−0.757	0.451	−0.030	to	0.013	−0.784	0.435	−0.076	to	0.033
MONO (×10^9^/L)	−1.456	0.149	−0.491	to	0.076	0.210	0.834	−0.755	to	0.934
NEUT (×10^9^/L)	−0.466	0.643	−0.032	to	0.020	−1.004	0.318	−0.115	to	0.038
LYMPH (×10^9^/L)	−0.402	0.688	−0.064	to	0.042	−0.132	0.896	−0.154	to	0.135
TG (mmol/L)	1.126	0.263	−0.010	to	0.036	0.753	0.453	−0.043	to	0.096
TC (mmol/L)	−0.692	0.490	−0.046	to	0.022	1.098	0.274	−0.039	to	0.136
HDL (mmol/L)	−1.362	0.176	−0.231	to	0.043	0.935	0.352	−0.144	to	0.401
LDL (mmol/L)	−2.640	0.010	−0.100	to	−0.014	0.144	0.886	−0.094	to	0.109
Apo A1 (g/L)	−2.072	0.041	−0.344	to	−0.007	0.662	0.512	−0.436	to	0.860
Apo B (g/L)	−1.652	0.102	−0.316	to	0.029	0.888	0.380	−0.411	to	1.052
Lpα (ng/L)	−0.237	0.813	0.000	to	0.000	0.182	0.857	−0.001	to	0.001
HCY (μmol/L)	−2.857	0.007	−0.009	to	−0.002	−0.377	0.707	−0.017	to	0.011
VB12 (pg/mL)	−0.808	0.427	0.000	to	0.000	0.222	0.827	−0.048	to	0.060
Folate (nmol/L)	−1.076	0.292	−0.028	to	0.009	1.167	0.255	−0.001	to	0.003

CRP, C-reactive protein; WBC, white blood cell; MONO, monocyte; NEUT, neutrophil; LYMPH, lymphocyte; TG, triglycerides; TC, total cholesterol; HDL, high-density lipoprotein; LDL, low-density lipoprotein; Apo A1, apolipoprotein A1; Apo B, apolipoprotein B; Lpα, lipoprotein a; H_cy_, homocysteine; VB12, vitamin B12. Bold indicates significant difference. CI, confidence interval.

## Data Availability

Not applicable.

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
