# Peer review of "Plasma Cell-Free DNA as a Novel Biomarker for the Diagnosis and Monitoring of Atherosclerosis"

_cells, 2022, doi:10.3390/cells11203248_

Round 1
Reviewer 1 Report
Abstract
"Plasma cell-free DNA (cfDNA), originating from dying cells, has been 17 recognized as a promising candidate biomarker…" – please rephrase the sentence since not only do dying cells release cfDNA. Similar to the first sentence of the introduction.
Methods
"cfDNA was then extracted from 200ml of serum using Se-83 rum/Plasma Circulating DNA Kit" – 200 ul instead of 200ml
"Eight-week-old male ApoE−/− mice were randomized in different experiment groups and blindingly fed a ND or WD for 2 and 6 months" – Please add more detail about ND and WD. I guess it means Normal died and Western Diet.
RESULTS:
Please add to the supplemental material a patient characteristics table.
"There is no association between cfDI and gender in AS patients; On the other hand, decreased cfDI levels were found in males in the control group (Figure S1C, D)" To support these data, patients' characteristics by gender should be well described.
It is a too strong conclusion that AUC 0.58 and 0.66 "validated the diagnosis value of cfDNA". It should be more discussed comparing to other biomarkers in the literature
“As displayed in Figure 2A and B, cfDI was inversely correlated with both carotid 169 plaque size (r = −0.285, p = 0.004) and thickness (r = −0.333, p = 0.001) (Figure 2A, B).” – Please add that this is a weak correlation.
Please explain why different primers were used in the human and animal models. ALU115 and ALU247 vs LINE1 5UTR and LINE1 ORF. For example, what is the amplicon size for both LINE1 5UTR and LINE1 ORF?
A short fragment (ALU 115) and a long fragment (ALU 247) were amplified and quantified for human targets. As for mouse targets, we used LINE1 OFR2 as the short fragment and LINE1 5'UTR as the long fragment – However, in the results section, the human data is expressed as the ratio of long/short, while in the animal model, it is expressed as the opposite. So it means that while the cfDI is decreasing in humans, it is actually increasing in the animal model.
Idem for figure 3F. If the LINE1 ORF2/LINE1 5'UTR ratio decreases, doesn't it mean that long fragments are increasing related to short fragments?
Discussion:
Since the main focus of the analysis of the study was the use of cfDNA from different fragments, it was expected in the discussion a more significant reference to other studies that performed this analysis in other models.
The discussion should also include a comparison of the AUC for the cfDNA values ​​found in the study with other studies that have already used other biomarkers.
Author Response
Response to Reviewer 1 Comments
1."Plasma cell-free DNA (cfDNA), originating from dying cells, has been recognized as a promising candidate biomarker…" – please rephrase the sentence since not only do dying cells release cfDNA. Similar to the first sentence of the introduction.
Response: We agree with the reviewer and have rephrased the sentence in the abstract section as follows (Page 1, lines 19-21):
“Plasma cell-free DNA (cfDNA), mainly originating from apoptosis, necrosis, and active secretion, has been recognized as a promising candidate noninvasive biomarker for diagnosis and prognosis of multiple diseases.”
The sentence was also revised in the introduction as follows (Page 1, lines 32-33):
“Circulating cell-free DNA (cfDNA) enter the peripheral blood mainly upon cell apoptosis, necrosis, and active secretion as part of regular cell turnover or pathology.”
2"cfDNA was then extracted from 200ml of serum using Serum/Plasma Circulating DNA Kit" – 200 ul instead of 200ml
Response: We appreciate your careful reading and apologize for the typo, and it has been corrected in the methods section (Page 2, line 95).
- "Eight-week-old male ApoE−/− mice were randomized in different experiment groups and blindingly fed ND or WD for 2 and 6 months" – Please add more detail about ND and WD. I guess it means a Normal Diet and Western Diet.
Response: We thank your suggestions and have added the details “normal diet (ND) or western diet (WD)” in the methods section (Page3, line140).
- Please add to the supplemental material patient characteristics table.
Response: We appreciate your insightful suggestion and have added the detailed characteristics for all patients and controls in Table S1 as follows:
Characteristics |
Controls (n=113) |
Atherosclerosis (n=102) |
P value |
Female (%) |
56 (49.6%) |
50 (49.0%) |
|
Age (year) |
54.8 ± 1.02 |
57.58 ± 1.20 |
0.1698 |
CRP (mg/dl) |
1.57 ± 0.37 |
6.38 ± 1.80 |
0.02 |
WBC (*10^9/L) |
5.88 ± 0.13 |
6.55 ± 0.18 |
0.0029 |
MONO (*10^9/L) |
0.29 ± 0.01 |
0.45 ± 0.01 |
<0.0001 |
NEUT (*10^9/L) |
3.5 ± 0.09 |
3.97 ± 0.16 |
0.0109 |
LYMPH (*10^9/L) |
1.82 ± 0.05 |
1.98 ± 0.08 |
0.0699 |
TG (mmol/L) |
1.57 ± 0.11 |
2.08 ± 0.17 |
0.0147 |
TC (mmol/L) |
4.89 ± 0.08 |
4.34 ± 0.13 |
0.0002 |
HDL (mmol/L) |
1.26 ± 0.03 |
1.42 ± 0.10 |
0.1594 |
LDL (mmol/L) |
2.85 ± 0.07 |
2.71 ± 0.10 |
0.2643 |
Apo A1 (g/L) |
1.53 ± 0.04 |
1.69 ± 0.10 |
0.3138 |
Apo B (g/L) |
0.91 ± 0.03 |
1.20 ± 0.12 |
0.1231 |
Lpα (ng/L) |
137.9 ± 25.45 |
246.4 ± 22.6 |
0.0075 |
HCY(μmol/L) |
13.55 ± 0.64 |
12.67 ± 1.99 |
0.6022 |
VB12(pg/mL) |
322.6 ± 21.32 |
501.5 ± 88.39 |
0.1005 |
Folate(nmol/L) |
11.2 ± 0.91 |
8.67 ± 0.89 |
0.0598 |
CRP, C reactive protein; WBC, white blood cell; MONO, monocyte; NEUT, neutrophil; LYMPH, lymphocyte; TG, triglycerides; TC, total cholesterol; HDL, high-density lipoprotein; LDL, low-density lipoprotein; Apo A1, apolipoprotein A; Apo B, apolipoprotein B; Lpα, Lipoprotein a; Hcy, homocysteine; VB12, vitamin B12. P value was calculated by student-t-test.
- "There is no association between cfDI and gender in AS patients; On the other hand, decreased cfDI levels were found in males in the control group (Figure S1C, D)" To support these data, patients' characteristics by gender should be well described.
Response: We agree with your concerns. The gender characteristics of all the participants have been presented in the new Table S1. Gender is an independent risk factor in atherosclerosis. Different life environments, genetics, and hormone backgrounds are involved in the development of AS in females [1]. While, other risk factors, such as LDL and flow shear could also contribute to the decrease in cfDI in female AS patients.
- It is a too strong conclusion that AUC 0.58 and 0.66 "validated the diagnosis value of cfDNA". It should be more discussed comparing to other biomarkers in the literature.
Response: We appreciate and agree with your suggestions. Increasing evidence demonstrated cfDI acts as a better biomarker than cfDNA concentration [2, 3], which is accordant with our current study. The conclusion about the diagnosis value of cfDNA has been revised as follows (Page 4, lines 169-171):
“Consistently, the ROC assay suggested that all three features of cfDNA could distinguish AS patients from normal controls (Figure 1D). Notably, cfDI showed more diagnostic potential for AS patients (AUC=0.780, p<0.0001).”
On the other hand, we have compared the AUC for cfDNA with other biomarkers in the discussion (Page 9, lines 277-286).
“Significant advances have been made in the novel biomarker identification for AS development and progression. A developed logistic model based on the well-known AS risk factors such as gender, age, hypertension, and diabetes shows an excellent predictive ability (AUC=0.808) and might have a significant clinical implication for detecting subclinical atherosclerosis in patients with chronic kidney disease [4]. Additionally, circulating miRNA-933 has been proven to be related to hyperlipidemia (AUC= 0.739) and atherosclerosis (AUC=0.703) [5]. A circulating protein panel including IGHA2, APOA, and HPT was also identified to predict AS development (AUC=0.73) [6]. Therefore, cfDI (AUC=0.780) identified in our study has a comparable diagnostic value as above biomarkers for AS patients. “
- “As displayed in Figure 2A and B, cfDI was inversely correlated with both carotid 169 plaque size (r = −0.285, p = 0.004) and thickness (r = −0.333, p = 0.001) (Figure 2A, B).” – Please add that this is a weak correlation.
Response: We have corrected the expression in the result (Page 5 line 181-183) as follows:
“cfDI showed a mild, but significant negative correlation with carotid plaque size (r = −0.285, p = 0.004) and thickness (r = −0.333, p = 0.001). “
- Please explain why different primers were used in the human and animal models. ALU115 and ALU247 vsLINE1 5UTR and LINE1 ORF. For example, what is the amplicon size for both LINE1 5UTR and LINE1 ORF?
Response: We agree with your concerns. In human samples, the ratio of ALU 115 and ALU 247 was most commonly used as the integrity of cfDNA [2, 7]. However, the human ALU primers could not be used in animal samples owing to the different genomics backgrounds. Alternatively, we chose LINE1 5UTR (amplicon size: 247 bp) and LINE1 ORF (amplicon size: 170 bp) to represent cfDNA and cfDI in the mouse models, which is widely used in previous studies for animal targets [8, 9] (Revised version Line 42-48 and Line 103-104).
- A short fragment (ALU 115) and a long fragment (ALU 247) were amplified and quantified for human targets. As for mouse targets, we used LINE1 OFR2as the short fragment and LINE1 5'UTR as the long fragment – However, in the results section, the human data is expressed as the ratio of long/short, while in the animal model, it is expressed as the opposite. So it means that while the cfDI is decreasing in humans, it is actually increasing in the animal model.
Response: We appreciate your careful reading and pointing out this problem. In the result part, the data should be Line1 5’UTR/ORF2 (long/short). We have carefully checked it and corrected all similar mistakes in the manuscript.
- Idem for figure 3F. If the LINE1 ORF2/LINE1 5'UTR ratio decreases, doesn't it mean that long fragments are increasing related to short fragments?
Response: Yes, the Y-axis should be labeled as Line1 5’UTR/ORF2 in Figure 3F. We apologize for this mistake and have corrected it in the revised manuscript.
- Since the main focus of the analysis of the study was the use of cfDNA from different fragments, it was expected in the discussion a more significant reference to other studies that performed this analysis in other models.
Response: We thank your enlightening suggestions and have expanded the discussion with more references (Page 9 line 257-266).
“Cell-free DNA (cfDNA) are DNA fragments of variable length between 50 and 250 base pairs which dissociated in the plasma [10]. cfDNA has become an attractive research subject as a noninvasive biomarker in diseases. As the index of cfDNA and cfDI, DNA fragments ALU 115 and ALU 247 have been explored in most diseases, such as infectious diseases, autoimmune disorders, myocardial infarction, and multiple cancers [11-14]. In our current study, we firstly found that cfDI could act as a novel biomarker during AS development and progression. Overall, our result and previous evidence indicated that cfDI could be used as a potential independent biomarker in most invasive diagnostics, improving disease monitoring, clinical decision, and patients’ outcome in the future.”
- The discussion should also include a comparison of the AUC for the cfDNA values found in the study with other studies that have already used other biomarkers.
Response: We appreciate your enlightening suggestion. We have compared the AUC for cfDNA with other biomarkers in the discussion (Page 9, lines 277-286).
“Significant advances have been made in the novel biomarker identification for AS development and progression. A developed logistic model based on the well-known AS risk factors such as gender, age, hypertension, and diabetes shows an excellent predictive ability (AUC=0.808) and might have a significant clinical implication for detecting subclinical atherosclerosis in patients with chronic kidney disease [4]. Additionally, circulating miRNA-933 has been proven to be related to hyperlipidemia (AUC= 0.739) and atherosclerosis (AUC=0.703) [5]. A circulating protein panel including IGHA2, APOA, and HPT was also identified to predict AS development (AUC=0.73) [6]. Therefore, cfDI (AUC=0.780) identified in our study has a comparable diagnostic value as above biomarkers for AS patients. “
Reference:
- Gasbarrino, K., D. Di Iorio, and S.S. Daskalopoulou, Importance of sex and gender in ischaemic stroke and carotid atherosclerotic disease. Eur Heart J, 2022. 43(6): p. 460-473.
- Hao, T.B., et al., Circulating cell-free DNA in serum as a biomarker for diagnosis and prognostic prediction of colorectal cancer. Br J Cancer, 2014. 111(8): p. 1482-9.
- Lamminaho, M., et al., High Cell-Free DNA Integrity Is Associated with Poor Breast Cancer Survival. Cancers (Basel), 2021. 13(18).
- Xiong, J., et al., A Nomogram for Identifying Subclinical Atherosclerosis in Chronic Kidney Disease. Clin Interv Aging, 2021. 16: p. 1303-1313.
- Xu, J., et al., Several circulating miRNAs related to hyperlipidemia and atherosclerotic cardiovascular diseases. Lipids Health Dis, 2019. 18(1): p. 104.
- Nunez, E., et al., Unbiased plasma proteomics discovery of biomarkers for improved detection of subclinical atherosclerosis. EBioMedicine, 2022. 76: p. 103874.
- Hussein, N.A., S.N. Mohamed, and M.A. Ahmed, Plasma ALU-247, ALU-115, and cfDNA Integrity as Diagnostic and Prognostic Biomarkers for Breast Cancer. Appl Biochem Biotechnol, 2019. 187(3): p. 1028-1045.
- Muotri, A.R., et al., L1 retrotransposition in neurons is modulated by MeCP2. Nature, 2010. 468(7322): p. 443-6.
- Kuroki, R., et al., Establishment of Quantitative PCR Assays for Active Long Interspersed Nuclear Element-1 Subfamilies in Mice and Applications to the Analysis of Aging-Associated Retrotransposition. Front Genet, 2020. 11: p. 519206.
- Al Sulaiman, D., et al., Length-Dependent, Single-Molecule Analysis of Short Double-Stranded DNA Fragments through Hydrogel-Filled Nanopores: A Potential Tool for Size Profiling Cell-Free DNA. ACS Appl Mater Interfaces, 2021. 13(23): p. 26673-26681.
- Ranucci, R., Cell-Free DNA: Applications in Different Diseases. Methods Mol Biol, 2019. 1909: p. 3-12.
- Hou, Y.Q., et al., Branched DNA-based Alu quantitative assay for cell-free plasma DNA levels in patients with sepsis or systemic inflammatory response syndrome. J Crit Care, 2016. 31(1): p. 90-5.
- Mondelo-Macia, P., et al., Circulating Free DNA and Its Emerging Role in Autoimmune Diseases. J Pers Med, 2021. 11(2).
- Ilatovskaya, D.V. and K.Y. DeLeon-Pennell, An Offer We Cannot Refuse: Cell-Free DNA as a Novel Biomarker of Myocardial Infarction. Am J Med Sci, 2018. 356(2): p. 88-89.
Reviewer 2 Report
The manuscript, entitled “Plasma cell-free DNA as a novel biomarker for the diagnosis and monitoring of atherosclerosis”, identified two promising biomarkers, circulating cell-free DNA (cfDNA) and DNA integrity of cfDNA (cfDI), for the diagnosis of atherosclerosis plaque progression. The authors have demonstrated that in atherosclerosis patients, the circulating level of cfDNA was increased and the cfDI was decreased compared to healthy normal people. This study also demonstrated the association of cfDI with the level of serum LDL, apolipoprotein A1, and homocysteine. Also, the authors demonstrated the profile of cfDNA and cfDI in different stages of plaque progression in a western diet-fed atherosclerosis mouse model. cfDI were negatively associated with carotid plaque size and thickness. Therefore, the study has a high impact on diagnosing atherosclerosis. However, there are some specific issues suggested for attention:
1. In the introduction, the author should add a brief description and rationale use of human targets DNA fragments ALU-115, and ALU 247 and mouse targets LINE1 OFR2 and LINE1 5’ UTR for assessment of serum levels of cfDNA.
2. In the section “sample preparation and extraction of cfDNA”, please check the amount of extracted cfDNA from 4ml venous blood… “cfDNA was then extracted from 200 ml of serum using Serum/Plasma Circulating DNA Kit (TIANGEN, BEIJING) following the manufacturer’s instructions”.
Author Response
Response to Reviewer 2 Comments
- In the introduction, the author should add a brief description and rationale use of human targets DNA fragments ALU-115, and ALU 247 and mouse targets LINE1 OFR2 and LINE1 5’ UTR for assessment of serum levels of cfDNA.
Response: We appreciate your enlightening suggestions and have expanded the introduction about the background of cfDNA. The revised introduction was presented as follows (Lines 42-48):
“As the size of plasma DNA can be attributed to the caspase-dependent cleavage, nucleosomal organization, and chromatin accessibility [1], the fragmentation profiles of cfDNA are involved with gene expression and multiple other cellular processes [2]. Therefore, DNA integrity of cfDNA (cfDI), which is a measure of the extent of cfDNA fragmentation, has been exploited as a noninvasive biomarker for diagnosis and prognostication in multiple diseases [3-5]. As the index of cfDNA and cfDI, DNA fragments ALU (ALU 115, short fragment and ALU 247, long fragment) and Line1 (Line1-OFR2, short fragment, and Line1-5’UTR, long fragment) have been explored in most diseases, such as infectious diseases, autoimmune disorders, myocardial infarction and multiple cancers [11-14]. Specifically, the ratio of ALU 115 and ALU 247 was commonly used as cfDI for human targets [3, 6, 7], while Line1-OFR2 and Line1-5’UTR were more often used in mice [8, 9].”
- In the section “sample preparation and extraction of cfDNA”, please check the amount of extracted cfDNA from 4ml venous blood… “cfDNA was then extracted from 200 ml of serum using Serum/Plasma Circulating DNA Kit (TIANGEN, BEIJING) following the manufacturer’s instructions”.
Response: We thank you for your careful reading and apologize for this typo. We have corrected it to 200 µl of serum in the methods section (Line 95).
Reference:
- Lo, Y.M., et al., Maternal plasma DNA sequencing reveals the genome-wide genetic and mutational profile of the fetus. Sci Transl Med, 2010. 2(61): p. 61ra91.
- Ulz, P., et al., Inferring expressed genes by whole-genome sequencing of plasma DNA. Nat Genet, 2016. 48(10): p. 1273-8.
- Lamminaho, M., et al., High Cell-Free DNA Integrity Is Associated with Poor Breast Cancer Survival. Cancers (Basel), 2021. 13(18).
- Higazi, A.M., et al., Diagnostic Role of Cell-free DNA Integrity in Thyroid Cancer Particularly for Bethesda IV Cytology. Endocr Pract, 2021. 27(7): p. 673-681.
- Stamenkovic, S., et al., Circulating cell-free DNA variables as marker of ovarian cancer patients: A pilot study. Cancer Biomark, 2020. 28(2): p. 159-167.
- Hao, T.B., et al., Circulating cell-free DNA in serum as a biomarker for diagnosis and prognostic prediction of colorectal cancer. Br J Cancer, 2014. 111(8): p. 1482-9.
- Hussein, N.A., S.N. Mohamed, and M.A. Ahmed, Plasma ALU-247, ALU-115, and cfDNA Integrity as Diagnostic and Prognostic Biomarkers for Breast Cancer. Appl Biochem Biotechnol, 2019. 187(3): p. 1028-1045.
- Muotri, A.R., et al., L1 retrotransposition in neurons is modulated by MeCP2. Nature, 2010. 468(7322): p. 443-6.
- Kuroki, R., et al., Establishment of Quantitative PCR Assays for Active Long Interspersed Nuclear Element-1 Subfamilies in Mice and Applications to the Analysis of Aging-Associated Retrotransposition. Front Genet, 2020. 11: p. 519206.
- Al Sulaiman, D., et al., Length-Dependent, Single-Molecule Analysis of Short Double-Stranded DNA Fragments through Hydrogel-Filled Nanopores: A Potential Tool for Size Profiling Cell-Free DNA. ACS Appl Mater Interfaces, 2021. 13(23): p. 26673-26681.
- Ranucci, R., Cell-Free DNA: Applications in Different Diseases. Methods Mol Biol, 2019. 1909: p. 3-12.
- Hou, Y.Q., et al., Branched DNA-based Alu quantitative assay for cell-free plasma DNA levels in patients with sepsis or systemic inflammatory response syndrome. J Crit Care, 2016. 31(1): p. 90-5.
- Mondelo-Macia, P., et al., Circulating Free DNA and Its Emerging Role in Autoimmune Diseases. J Pers Med, 2021. 11(2).
- Ilatovskaya, D.V. and K.Y. DeLeon-Pennell, An Offer We Cannot Refuse: Cell-Free DNA as a Novel Biomarker of Myocardial Infarction. Am J Med Sci, 2018. 356(2): p. 88-89.
Round 2
Reviewer 1 Report
The authors made several adjustments to the article that improved the points raised. However, it is still necessary a well-detailed description of the different diets used in animal models
Author Response
The authors made several adjustments to the article that improved the points raised. However, it is still necessary a well-detailed description of the different diets used in animal models
Response: Thanks to your positive comments and suggestion, we have revised the method as follows (Page 3, line 135-138):
“The composition of normal chow diet is based on the NIH feeds standard (NIH-07). The composition of normal chow diet is based on the NIH feeds standard (NIH-07). The western diet was obtained from the Research Diets (D12079B, New Brunswick, NJ) containing 17 kcal% protein, 43% carbohydrate, and 41 kcal% fat.”